# Explainable Artificial Intelligence for Predicting Attention Deficit Hyperactivity Disorder in Children and Adults

**DOI:** 10.3390/healthcare13020155

**Published:** 2025-01-15

**Authors:** Zineb Namasse, Mohamed Tabaa, Zineb Hidila, Samar Mouchawrab

**Affiliations:** 1Multidisciplinary Laboratory of Research and Innovation, Moroccan School of Engineering Sciences, Casablanca 20250, Morocco; z.namasse@emsi.ma (Z.N.); z.hidila@emsi.ma (Z.H.); 2Research, Development, and Innovation Laboratory, Mundiapolis University, Casablanca 20180, Morocco; s.mouchawrab@mundiapolis.ma

**Keywords:** Attention Deficit Hyperactivity Disorder, Machine Learning, Explainable Artificial Intelligence, children, adults, anxiety, depression, bipolar disorders

## Abstract

Attention Deficit Hyperactivity Disorder (ADHD) is a disorder that starts in childhood, sometimes persisting into adulthood. It puts a strain on their social, professional, family, and environmental lives, which can exacerbate disorders such as anxiety, depression, and bipolar disorder. **Background/Objectives**: This paper aims to predict ADHD in children and adults and explain the main factors impacting this disorder. **Methods:** We start by introducing the main symptoms and challenges ADHD poses for children and adults such as epilepsy and depression. Then, we present the results of existing research on three ADHD comorbidities: anxiety, depression, and bipolar disorder, and their possible continuity in adulthood with therapeutic implications. After that, we explain the impact of this disorder and its relationship with these comorbidities on the affected patient’s health and environment and list proposed treatments. We propose a methodology for predicting this impairment in children and adults by using Machine Learning algorithms (ML), Explainable Artificial Intelligence (XAI), and two datasets, the National Survey for Children’s Health (NSCH) (2022) for the children and the ADHD|Mental Health for the adults. **Results:** Logistic Regression (LR) was the most suitable algorithm for children, with an accuracy of 99%. As for adults, the XGBoost (XGB) was the most performant ML method, with an accuracy of 100%. **Conclusions:** Lack of sleep and excessive smiling/laughing are among the factors having an impact on ADHD for children, whereas anxiety and depression affect ADHD adults.

## 1. Introduction

Mental disorders represent a significant public health issue, impacting millions of people around the world. The World Health Organization (WHO) [1] stated that their prevalence rate varies from 10.9% in the World Health Organization African Region to 15.6% in the World Health Organization Americas Region. Mental disorders are slightly more common in high-income countries (15.1%) but are also common in low-income countries (11.6%); among them is the Attention Deficit Hyperactivity Disorder (ADHD).

ADHD is a neurodevelopmental disorder that affects 3–10% of children and 4–5% of adults (Desseilles et al. [2]). The main symptoms are Attention Deficit and hyperactivity/impulsivity disorder (Simmons RW et al. [3] and the Diagnostic and Statistical Manual of Mental Disorders, 5th edition (DSM-V)). Martella D et al. [4] stated several theories explaining these symptoms, but two theoretical models are the most predominant: the inhibitory and cognitive-energetic models. First, the inhibitory model posits that the primary deficit in ADHD is poor behavioral inhibition related to the execution network specific to the attention system. It involves three interdependent processes: (1) preventing an initial response from dominating an event, (2) ending a response in progress, and (3) maintaining control of the interference of divergent information. Poorly functioning inhibition can damage the effectiveness of executive functions (such as non-verbal working memory, motivation, self-regulation of affect, etc.). Second, the cognitive-energetic model presents energy factors as the most critical explanation of ADHD symptoms, suggesting that deficits in these factors lead to symptoms of inattention and hyperactivity/impulsivity. This model maintains that the overall efficiency of the information process is defined by the exchange between three mechanisms at different levels, following ascending and descending flows. The lower level contains four global stages of attentional calculation mechanisms: encoding, retrieval, decision, and motor organization. The intermediate level is made up of the following three energy pools: arousal, effort, and activation. The upper level designates the executive control system and includes many abilities, namely planning, response prevention, error identification and correction, working memory, and others. Conforming to the Centers for Disease and Prevention Data and Statistics on ADHD [5], it occurs more often in boys (15%) than girls (8%). Desseilles et al. [2] mentioned that ADHD boys are more impulsive, while girls tend to be more inattentive. If we refer to the World Mental Health Report [1], among the mental health problems that cause mental disorders such as depression, ADHD, and bipolar disorder are genetic factors, low level of education, obesity, and other metabolic risks, chronic diseases, sleep disorders, bullying, discrimination and social exclusion, pollution vitamin D deficiency, and so on. In this paper, we present the following sections: Section 2 presents in depth the challenges faced by children and adults with ADHD. Section 3 explains the symptoms, the prevalence, the link between ADHD and three of its comorbidities: anxiety, depression, and bipolar disorder, their continuity into adulthood, and the proposed therapeutic implications. Section 4 discusses the impact of ADHD on the health and the environment of the patient affected by ADHD and treatments such as dietary intake. Section 5 proposes a methodology with two datasets, ML methods, and an introduction to Explainable AI. Section 6 provides results and discussion, and Section 7 concludes. This paper explores how Explainable AI and Machine Learning techniques can contribute to the prediction of ADHD and the understanding of associated comorbidities in children and adults.

## 2. Background

ADHD’s reactions may differ between children and adults. For example, children tend to fidget with their hands and feet, get up in class without permission, climb and run at every opportunity, and find it challenging to stay still where they are supposed to be quiet. In contrast, adults have an inner need to move, are unable to relax, answer questions before obtaining permission from the other person, have difficulties waiting in a queue, impose their presence at inappropriate moments, and so on (Desseilles M et al. [2]). Many authors also identified significant challenges faced by ADHD children, such as sleep disorders (Kim WP et al. [6]), a very high step rate (Meachon EJ et al. [7]), proactive and/or reactive aggression (Ayasrah MN et al. [8]), epilepsy (Park KJ et al. [9]), anxiety (Gair SL et al. [10]) and many other obstacles. Other studies pointed out difficulties ADHD adults struggle with. For example, high caffeine consumption (Cohen A et al. [11]), lack of punctuality and diligence at work (Durand G et al. [12]), alcohol dependence (Pierre-paul L [13]), bipolar and mood disorders (Mishra VC et al. [14]), pragmatic disorders (Even-Simkin E [15]), depression (Borden LA et al. [16], Powell V et al. [17]), anxiety (Alarachi A et al., Sacco R et al. [18,19]) and so forth.

## 3. Comorbidities, Relationships with ADHD, Continuity and Therapeutic Implications

### 3.1. Children

#### 3.1.1. Anxiety

Prevalence and symptoms: Anxiety is a common comorbidity associated with ADHD. Several authors reported anxiety rates in different countries such as England, Australia (nationwide), Turkey (Izmir) and Lithuania (nationwide), where anxiety had the highest prevalence at 7.9% (Sacco R et al. [19]), 7.7% to 32.7% among 6–12 years old children in Saudi Arabia (Alfakeh SA et al. [20]), 1.3% in Chinese schoolchildren aged 6–16 years (Wang F et al. [21]). Others focused on the anxiety symptoms prevalence, such as Mathur R [22], who found out that 43.08% of the children in Rajasthan, India, are agitated, 49.23% lack concentration, 55% are nervous, 18% lack sleep, 14% breath with difficulty, 18% were trembling and 9% felt weak.

Relationship with ADHD: Research has established hypotheses about ADHD associated with anxiety. For example, among the suggested theories by Koyuncu A et al. [23], the two disorders share the exact mechanism explained by a common heredity. Ref. [24] explained that genetic and environmental factors cause this occurrence. Another theory emphasized that ADHD could lead to a rise in comorbidity. Predominant inattention symptoms in ADHD patients were associated with social anxiety disorder (SAD).

Additionally, Okyar E et al. [25] deduced, from treatment-naive children and adolescents with somatic symptoms of ADHD and anxiety, that ADHD people are more anxious. Another assumption made by Wu M et al. [26] pointed out that young people with ADHD and autism are potentially at a greater risk of global anxiety, specifically generalized anxiety, separation anxiety, and social anxiety. It seemed that ADHD/ASD patients were more likely to suffer from anxiety than their peers.

#### 3.1.2. Depression

Prevalence and symptoms: Among anxiety, various studies sought to estimate depression prevalence, such as Chen J et al. [27], who found that 28.6% of Chinese children and adolescents were depressed during the COVID-19 pandemia, 27% of the children and adolescents suffered from depression with Type 1 Diabetes (T1D) in Saudia Arabia (Alaqeel A et al. [28]), 20.9% of American children from 12 to 17 years old were experiencing major depression (Bitsko RH et al. [29]), 42.3% of Iranian children experienced a depressive episode (Mahmudi L et al. [30]). Others reported an increased rate of depression through adolescence. Maughan B. et al. [31] reviewed data on depression in children and adolescents using a narrative review. They concluded that the unipolar depression rate is lower than at puberty but increases during adolescence, particularly in girls. The consequences of depression during this period include suicide, social functioning problems, and poor physical and mental health.

Relationship with ADHD: In a specialist review carried out by Thapar A et al. [32], several theories could explain the relationship between ADHD/ASD and depression. For example, associations between disorders causing medication side-effects, a case study of current diseases, genetic-environmental risks resulting in both ADHD/ASD and depression or a third disorder occurring with neurodevelopmental disorders. Eng AG et al. [33] reported on possible gender differences in ADHD and puberty-related disorders and comorbidity. Depressive symptoms increased significantly with age and were higher in young people at later puberty. Subsequently, Gonzalez VJ et al. [34] sought to evaluate disorders such as anxiety, depression, and ADHD in people with congenital heart disease (CHD) compared with those without. 1164 had CHD. 18.2% of them were diagnosed with anxiety or depression, compared with 5.2% without CHD.

#### 3.1.3. Bipolar Disorders

Prevalence and symptoms: Bipolar Disorder is an impairment where the affected patient moves drastically from paroxysm to depression (Wu Y et al. [35]). Many factors could lead to this disorder. One of which is family transmission, where 20% of bipolar disorder transmission risk occurs between generations. Several studies revealed bipolarity prevalences. For instance, Mohammadi MR et al. [36], with a prevalence of 0.26% for Iranian males and 0.29% for female Iranian children, 2.2% of US adolescents (Post RM et al. [37]). Others focused on the bipolar disorders symptoms, such as low-noise inflammation, internal clock (sleep) sleep disorders and stress, irritability, instability, lack of concentration, and increased goal-directed energy (Rantala MJ et al. [38] and Nebhinani N [39]).

Relationship with ADHD: Various researchers made attempts to test the link between bipolarity and ADHD. Among them, Pouchon A. et al. [40] observed that three double-blind, randomized, placebo-controlled trials (RCTs) examined how children and adolescents with bipolar disorder treated with a mood stabilizer responded to pharmacological treatments for ADHD, or Long Y et al. [41], who brought up, in a meta-analysis of voxel-based whole-brain morphometry studies on bipolar disorder and ADHD, that both disorders were associated with a decrease in the following brain regions: the right insula and the anterior cingulate cortex, both implicated in emotional regulation and attention.

### 3.2. Adults

#### 3.2.1. Anxiety

Continuity with ADHD: ADHD comorbidities, like anxiety, can persist in adulthood. Many studies have found symptoms among ADHD students, such as worry and more maladaptive beliefs (O’Rourke SR et al. [42]). Others pointed out anxiety prevalence in ADHD adults; for example, 46.6% of Saudi Arabian adults have ADHD (Alharbi N et al. [43]).

Therapeutic implications: Researchers found treatments to reduce anxiety in ADHD people. Among them, León-Barriera R et al. [44] confirmed that CBT yielded better results in both studies and that the children had a better quality of life. Hanssen KT et al. [45] concluded that goal management training (GMT) combined with psychoeducation exhibited much lower symptoms of anxiety than the TAU group.

#### 3.2.2. Depression

Continuity with ADHD: Numerous studies highlighted the pursuit of depression and ADHD in adulthood. For example, Powell V et al. [17] discovered that 12.8% of women suffering from persistent depression had ADHD, and 3.4% met the criteria for ADHD according to the DSM-5. Riglin L et al. [46] found out that ADHD in children was combined with a higher risk of persistent ADHD in young adults affected by this disorder. Furthermore, Ishizuka K et al. [47] examined whether self-perceptions of ADHD-related characteristics were consistent regardless of changes in the severity of depressive symptoms. It seemed that there was a strong relationship between a change in the BDI and ASRS results, and as the BDI increased, the frequency of positive ADHD screening by the ASRS decreased.

Therapeutic implications: Several researchers worked on therapies to alleviate depression in ADHD adults, such as Poissant H et al. [48] and MORAITI et al. [49], who found that treatments based on mindfulness were adequate and a good approach to reduce this comorbidity. Digital awareness also involves better behaviors, like improved memory and concentration abilities, for extended periods (MORAITI et al. [49]).

#### 3.2.3. Bipolar Disorders

Continuity with ADHD: Some studies focused on relationships between sociodemographic factors and bipolarity in ADHD adults. For instance, Bartoli F et al. [50] found that men with ADHD, unemployed and unmarried patients, were more prevalent and also exhibited episodes of bipolar disorder. Others on the association between bipolar disorders and ADHD. As an example, Schiweck C et al. [51], through a meta-analysis, found out that about one in thirteen adults affected by bipolar disorder also had ADHD.

Therapeutic implications: Many authors pointed out the importance of mood stabilization to reduce bipolarity in ADHD adults. Such as MacDonald L et al. [52], who noticed that mood stabilization should precede the initiation of treatment with psychostimulants for ADHD to reduce excessive manic effects, with antipsychotics and atypical antidepressants potentially being highlighted depending on clinical presentations, emphasizing the union between psychotherapy and pharmacological solutions. However, other studies mentioned complaints about this treatment because of its side effects, such as experiencing memory lapses or concentration problems (Salvi V et al. [53]).

## 4. Impact of ADHD on the Health and Environment of the Affected Patient and Some Treatments

### 4.1. Health

Li L et al. [54] wanted to establish a correlation between ADHD symptoms and the consumed food. They found that inattention is positively correlated with the consumption of seafood, nutrient-rich foods high in fats and sugars, and unhealthy foods, ranging from 0.03 (95% CI: 0.01, 0.05) for seafood to 0.13 (95% CI: 0.11, 0.15) for sugary foods. It is also negatively correlated with the fruits and vegetables consumption and the profile of healthy foods at −0.06 (95% CI: −0.08, −0.04) for vegetables and nutritious foods at −0.07 (95% CI: −0.09, −0.05) for fruits, and hyperactivity/impulsivity has similar but less correlated associations of 0.03 (95% CI: 0.01, 0.05) for high-fat foods and 0.09 (95% CI: 0.06, 0.11) for high-sugar foods, along with negative correlations ranging from −0.02 (95% CI: −0.04, −0.01) for vegetables to −0.03 (95% CI: −0.05, −0.01) for fruits. Furthermore, Xie Y et al. [55] assessed the effectiveness of physical activity (PA) in patients with ADHD. They found that among the nine before-and-after studies with 232 participants and the 14 controlled studies with two groups (162 participants/141 non-ADHD), the combined results demonstrated that AP significantly improved all symptoms of ADHD (inattention: SMD = 0.604, 95% CI: 0.374 to 0.834, *p* < 0.001; hyperactivity/impulsivity: SMD = 0.676, 95% CI: 0.401 to 0.950, *p* < 0.001; emotional problems: SMD = 0.416, 95% CI: 0.283 to 0.549, *p* < 0.001; behavioral problems: SMD = 0.347, 95% CI: 0.202–0.492, *p* < 0.001). Salvat H et al. [56] aimed to assess the nutrient intake, dietary habits, and anthropometric variables in children affected by ADHD compared to children without ADHD. It turned out that children with ADHD consumed more simple sugars, tea, and prepared food but ate less protein, vitamin B1, vitamin B2, vitamin C, zinc, and calcium, which resulted in a higher body mass index (BMI) compared to children without ADHD. In addition, Yatzkar U et al. [57] focused on the effects of 6 months of omega-3 fatty acid enrichment on the fatty acid profile of erythrocytes and the clinical severity of ADHD symptoms in children. Their results showed that omega-3 fatty acid supplements in children with ADHD significantly increased their omega-3 index, rising from an average of 4.4% at baseline to 11.6% after 6 months, and positively affected ADHD symptoms. Baniasadi TM [58] aimed to study the effects of participation in physical activity and exercise on social and adaptive functioning in children with ADHD. Rucklidge JJ et al. [59] studied the efficacy and safety of a foreign micronutrient formula, primarily composed of vitamins and minerals, without omega fatty acids, in treating adults with ADHD. Among their results, individuals receiving micronutrients showed a more significant improvement than those receiving a placebo, both overall and in ADHD symptoms.

### 4.2. Environment

According to Wymbs BT et al. [60], romantic relationships for ADHD patients do not last for long, do not experience them for long enough, and have more conflicts than adults without ADHD. These conflicts within individuals with physical and mental health issues increase the risk of depression, substance abuse, and so on. Claussen AH et al. [61] also assessed the links between parenting and family environment factors with ADHD. They concluded that children’s education and family environment influence ADHD symptoms and can impact the lifestyle of an ADHD child. Furthermore, DOULOU et al. [62] proposed intervention forms to enhance the life quality of ADHD children, such as video games, virtual reality (VR), or augmented reality. (AR). They reported that the video game “EmoGalaxy” enhances the social skills of children with ADHD, such as recognizing, expressing, and stabilizing emotions. VR allows for immersive and controlled learning environments designed for individuals with ADHD. AR, for its part, associates virtual and fundamental elements for better environmental adaptation. Rosi E et al. [63] attempted to find evidence linking ADHD and environmental factors. They discovered that many pollutants, such as heavy metals, increase the risk of developing ADHD. In addition to that, Hollingdale J et al. [64] addressed the impact of COVID-19 on patients with ADHD. They concluded that the pandemic had exacerbated the symptoms and comorbidities of this group of people, such as behavioral and emotional disorders, irritability, oppositional disorders, isolation, depression, obsessive-compulsive disorders, sleep disturbances, and sedentary and overconsumption of digital devices, exposing them to blue light, as for Shabat T et al. [65] evaluated the environmental factors and the participation of children and young adults with or without ADHD in three contexts: at home, at school, and in the community. They noticed that the ADHD population had less favorable outcomes than the typical development group and that parents supported these children to improve their participation in these contexts. Finally, the review published between 1987 and 2022 conducted by Sin BSY et al. [66] examined the effectiveness of psychoeducation and adapted home therapy for children with ADHD. They concluded that parents play a fundamental role in supporting children with ADHD to self-regulate, reduce symptoms and potential comorbidities, overcome learning difficulties, and improve the parent–child relationship through psychoeducation.

## 5. Materials and Methods

### 5.1. Datasets

To better understand the behaviors of ADHD patients (children and adults), we have used two datasets:

#### 5.1.1. Dataset for Children

Here, we used the National Survey of Children’s Health (2022) (NSCH) Database [67,68]. It usually contains more than 50,000 rows, each representing a child aged 3 to 17. It also has more than 900 features, among which are those related to comorbidities, namely anxiety_22 and depression_22.

Preprocessing: The dataset first contained 933 features and 54,103 rows. We then selected features by manually removing erroneous or missing values represented by 99, features unrelated to children, like A1_DEPLSTAT/A2_DEPLSTAT, which concerns adults, or MOMAGE, representing the mother’s age. According to an article by Sepehrmanesh Z et al. [69] on the links between breastfeeding and ADHD, the mother’s age during pregnancy does not play a significant role in the cases of ADHD in comparison to non-ADHD cases. After this operation, we had a dataset with 39,924 candidates and 63 features. Furthermore, ADHD_22 initially contained three options: (1) The child does not have ADHD, (2) the child has heard that they might have ADHD but does not, and (3) the child has ADHD. A value of 95 indicates that the child is between 0 and 2 years old. After a manual transformation, the ADHD_22 had two choices: the values 1, 2, and 95 were replaced by 0 (non-ADHD), and the value three was replaced by 1. (ADHD).Split: We divided the dataset into training, validation, and test sets: 70% for training, 10% for validation, and 20% for testing with a random state of 42.

Some features of the dataset on children with ADHD are shown in Table 1:

#### 5.1.2. Dataset for Adults

As for the adults, we opted for the ADHD | Mental Health Dataset provided by the Kaggle platform [70]. The dataset contained 506 adults aged 18 to 22 and 84 features, including those related to impairments such as bdi1_total for depression or bai1_total for anxiety.

Preprocessing: The dataset had 84 features at the beginning. We proceeded as follows: We started by manually removing the incorrect or erroneous values represented by n/a, na..., and indexes representing only a part of the total number of an index, so we kept instead the total numbers (bai1_total, bdi1_total). After this processing, the dataset contained 354 candidates and 11 features (we added one feature for the prediction).Split: To split this dataset, we planned as follows: we divided it into a training set, a validation set, and a test set. 55% were dedicated to training, 5% to validation, and 40% to testing. The random_state value was 16.

A selection of the 11 features of the dataset on adult ADHD are described in Table 2:

### 5.2. Methods

ADHD can significantly affect the lives of individuals, whether they are children and/or adults, leading them to face circumstances that can impact their social, academic, and behavioral lives. These challenges can lead to isolation and a lower quality of life for patients with ADHD. To this end, we proposed the following methodology. We started by handling missing, erroneous values and features not related to ADHD in children and adults in both datasets. Then, we split them into training, validation, and testing sets. After that, we employed Machine Learning (ML) methods to predict ADHD in children and adults and Explainable Artificial Intelligence (XAI) to reveal the features that contributed, positively or negatively, to the ADHD prediction.

#### 5.2.1. Choice of Models

For the case of children, we used five models: the Logistic Regression (LR), the Decision Tree (DT), the K-nearest neighbors (KNN), the Support Vector Machine (SVM), and the MultiLayer Perceptron Classifier (MLPC). We chose these five algorithms, specifically LR over others, for their simplicity and a lower risk of overfitting. Furthermore, LR requires fewer computational resources than Deep Neural Networks (DNN). We have chosen the SVM, XGBoost (XGB), and the MLCP regarding adults. We chose these three options over other algorithms due to our small dataset of ADHD adults. Moreover, these methods, particularly XGB, are efficient, allowing for rapid training, unlike others, such as DNN, which takes a long time to train.

In this section, we will explore the algorithms used in great detail:

#### 5.2.2. Details of Models Used

Logistic Regression (LR): LR is an algorithm of classification based on supervised learning that aims to make predictions about dependent variables from independent variables. Referring to Cramer JS [73], the sigmoid function is:(1)$$P\left(z\right)=\frac{e^{z}}{1+e^{z}}$$

P acts as a symmetric density distribution function, like a midpoint of 0. When Z moves within the interval [−4, 4], P moves between 0 and 1. This function varies, depending on the meaning of the variables. In logistic regression, P has several variants and $Z=x^{t}+\beta$, where x is the vector of covariates (including a constant unit) and β is their coefficient. However, the logistic function essentially describes the evolution of a proportion P over time t, with Z = α + βt, since P(t) increases monotonically with t. It is an increasing curve.

The logistic function was created in the 19th century to describe population growth and the evolution of autocatalytic chemical reactions. In both cases, the temporal trajectory of a quantity W(t) and its growth rate are calculated as follows:(2)$$W(t)=\frac{dW(t)}{d(t)}$$

The simplest hypothesis is that *W(t)* is proportional to itself:(3)$$W(t)=\beta W(t),\beta =\frac{W(t)}{W(t)}$$
where *β* is the growth coefficient; this leads to exponential growth.(4)$$W(t)=Ae^{\beta t}$$
where A is often replaced by *W*.(0). This is a reasonable demographic growth model without opposition in a young country like the United States in its early years. “A population left to itself will increase in geometric progression” [74]. However, Alphonse Quetelet (1795–1874) knew that a blind extrapolation of exponential growth would lead to impossible results. Like Quetelet, Verhulst (1804–1849) analyzed the problem by adding a term to represent increasing resistance:(5)W(t)=βW(t)(∅ −W(t))

By experimenting with several variants of *∅*, the logistic function appears when it is quadratic, so it is written as follows:(6)W(t)=βW(t)(Ω -W(t))
where *Ω* is the upper limit of the saturation level of *W*; the growth is now proportional to both the already reached population *W(t)* and the remaining space to go further *Ω − W.(t)*. If *W(t)* is expressed as a proportion *P(t) = W(t)/Ω*, this gives the following:(7)P(t)=βP(t)(1−P(t))

The solution to this differential equation is as follows:(8)$$P\left(t\right)=\frac{e^{\alpha +\beta t}}{1+e^{\alpha +\beta t}}$$

Decision Tree (DT): According to Charbuty B et al. [75], DT is another classification algorithm that aims to establish a ranking based on homogeneous branches and nodes. There are numerous types of decision trees. Some of them are presented in Table 3:

Entropy is used to calculate the impurity or randomness of a dataset. It is between 0 and 1. The closer it is to 0, the better it is. It is computed as follows:(9)$$\mathrm{Entropy}(S)=\sum_{i=1}^{C}P_{i}\log2^{P_{i}}$$
where *Pi* is the ratio between the number of samples and the i-th subset of assigned values.

Gain is a measure used for segmentation and is often referred to as mutual information. It indicates the degree of knowledge of the value of a random variable. Unlike entropy, the higher the gain, the better it is. It is calculated using the following equation:(10)$$\mathrm{Gain}(S,A)=\sum_{v\in V(A)}\frac{\left|S_{v}\right|}{\left|S\right|}Entropy\left(S_{v}\right)$$
where the range of attribute A is V(A), and $S_{v}$ is a subset of the set *S* equal to the value of attribute *v*.

K-Nearest Neighbors (KNN): Conforming to Taunk K et al. [76], KNN is a classification algorithm based on supervised non-parametric learning. This algorithm provides a set of labeled data where the data points are classified into several categories so that the non-labeled data category is predicted. KNN was developed to analyze features where obvious approximations of parametric probability density were unavailable or difficult to define. In an unpublished article from the United States Air Force aviation school in 1951, Fix and Hodges introduced a non-parametric supervised algorithm called KNN. The KNN classification involves the learning phase (a developed classifier based on the training data) and the classifier evaluation.

The algorithm uses the variable K to identify better the category to which a class belongs. The steps are as follows: First, it analyzes the K nearest points to the new data point, that is, the K nearest neighbors. Secondly, with the help of the neighboring classes, KNN identifies the class into which the new data should be placed. Once we have gathered the K’s nearest neighbors, we need to make the most of them to predict the class of the learning instance. Here are the following steps using a dataset $\left(\{\left(x_{1},y_{1}\right),\left(x_{2},y_{2}\right),\ldots ,(x_{n},y_{n})\}\right)$:

Firstly, we need to store the training set. Secondly, for each non-labeled dataset, we compute the Euclidean distance to all training data points using the following formula:(11)$$D(x,y)=\sqrt{{\left(x_{1}-y_{1}\right)}^{2}+(x_{2}-y_{2})²+\ldots +(x_{n}-y_{n})²}$$

Next, we need to find the K nearest neighbors and choose the class that has the most K nearest neighbors.

After storing the training set, all parameters must be normalized to simplify the computation. The classification result depends on the K value. If K equals 1, the data are assigned to the nearest neighboring class.

Support Vector Machine (SVM): As BOSWELL et al. [77] mention, SVM is a classification and regression algorithm that seeks to find the most optimal hyperplane to better separate classes. The first published paper about SVM was in 1995 [78].

According to BOSWELL et al. [77], we have training examples *‘l’*
$\left\{x_{i},y_{i}\right\}$*, i =* 1, *...*, *l* where each example contains “*d*” inputs ($xi\in R^{d}$) and a class label with one of the two following values (*yi* ∈ {–1,1}). Currently, all hyperplanes in $R^{d}$ are parameterized by a vector *(w)* and a constant *b*, according to the following formula:(12)w*x+b=0

Based on the hyperplane *(w,b)* separating the data, this gives us the function below:(13)f(x)=sign(w*x+b)
that correctly classifies the trained data. Yet, the hyperplane represented by *(w,b)* is also expressed by all pairs *(λw,λb)* where $\lambda \;\epsilon \;R^{+}$. We, therefore, define a canonical hyperplane that separates the data from the hyperplane by a distance of at least 1. In other words,(14)$$y_{i}\left(x_{i}*w+b\right)\geq 1\forall i$$

All hyperplanes have a distance function greater than or equal to 1. (Do not confuse this with Euclidean distance). Based on a given hyperplane *(w,b)*, all pairs *(λw,λb)* define the same hyperplane, but each has a different distance function to a given point. To calculate the distance from the hyperplane to a given point, we normalize with the intensity of *w*. This is computed as follows:(15)$$d((w,b),x_{i})=\frac{y_{i}(x_{i}*w+b)}{\left||w|\right|}\geq \frac{1}{\left||w|\right|}$$

This equation shows the hyperplane that maximizes the geometric distance to the nearest data points, which has been successfully chosen by minimizing||*w*||. The primary method is Lagrange multipliers [79]. The problem turns into the following:(16)$$\mathrm{minimize}:\;W(\alpha )=-\sum_{i=1}^{l}\alpha_{i}+\frac{1}{2}\sum_{i=1}^{l}\sum_{j=1}^{l}y_{i}y_{j}\alpha_{i}\alpha_{j}(x_{i}.x_{j})$$(17)$$\mathrm{subject}\;\mathrm{to}:\;\sum_{i=1}^{l}y_{i}\alpha_{i}=0\leq \alpha_{i}\leq C(\forall i)$$
where *α* is the vector of non-negative Lagrange multipliers *λ* to be determined, and C is a constant.

XGBoost (XGB): Before discussing XGB, we will explain Gradient Boosting. By Bentéjac C et al. [80], boosting algorithms combine weak models that learn slightly better than random models into a strong model iteratively (Robert E. Schapire Yoav Freund [81]). Gradient boosting is a regression-boosting algorithm (Jerome H. Friedman [82]). Based on a dataset $D={\left\{x_{i},y_{i}\right\}}_{N}^{1}$, Gradient Boosting aims to approximate $\hat{F}(x)$ to the function *F*(x)* that connects the instances of x to their final values y by decreasing the expected value of a given loss function L(y, F(x)). Gradient Boosting implements an additional approximation of *F*(x)* in a weighted sum of functions:(18)$$F_{m}\left(x\right)=F_{m-1}\left(x\right)+\rho_{m}h_{m}(x)$$
where $\rho_{m}$ is the weight of the m-th function, $h_{m}(x)$. These functions are the set models (usually the decision tree). First of all, a constant approximation of the function *F*(x)* is calculated as follows:(19)$$F_{0}\left(x\right)={argmin}_{\alpha }\sum_{i=1}^{N}L(y_{i},\alpha )$$

The following models are typically minimized:(20)$$\left(\rho_{m},h_{m}\left(x\right)\right)={argmin}_{\rho ,h}\sum_{i=1}^{N}L(y_{i},F_{m-1}\left(x_{i}\right)+\rho_{h}(x_{i}))$$

XGB is a classification and regression algorithm that relies on Gradient Boosting and the ensemble of decision tree algorithms (Chen T et al. [83]) to be highly scalable. Like Gradient Boosting, it realizes an extension of the objective function by minimizing the loss. The loss function, according to Bentéjac C et al. [80], is calculated via(21)$$L_{xgb}=\sum_{i=1}^{N}L\left(y_{i},F\left(x_{i}\right)\right)+\sum_{m=1}^{M}Ω(h_{m})$$
where Ω is the regularization function to avoid or reduce the overfitting, computed by the following equation:(22)$$Ω(h)=\gamma T+\frac{1}{2}\lambda |\left|w\right||²$$

Multi-Layer Perceptron Classifier (MLPC): MLPC is an algorithm based on ANN and, in compliance with Pal SK et al. [84], involves several layers of processing elements (nodes) or simple two-state sigmoid neurons responding to each other through weighted connections. After a lower input layer, there are usually hidden layers followed by an output layer. There is no interconnection within a layer, while all the neurons are fully connected to the adjacent layers. The weights represent the correlation levels between the activity levels of the neurons they connect.

The total input layer received by neuron j in layer h+1 is determined as follows:(23)$$x_{j}^{h+1}=\sum_{i}y_{i}^{h}w_{ji}^{h}-\theta_{j}^{h+1}$$
where $y_{i}^{h}$ is the state of the i-th neuron in layer *h* and $w_{ji}^{h}$ its weight from the i-th neuron in layer *h* to the j-th neuron in layer *h*+1, and $\theta_{j}^{h+1}$ is the threshold of the j-th neuron in layer *h*+1. For a given layer, the output of a neuron (except the input layer) is a non-linear monotonic function of its total input and is given as follows:(24)$$y_{j}^{h}=\frac{1}{1+e^{{-x}_{j}^{h}}}$$

For the input layer nodes,(25)$$y_{j}^{0}=x_{j}^{0}$$

#### 5.2.3. Explainable Artificial Intelligence and SHAP

In line with the UNESCO General Conference, during a meeting in Paris between 9 November and 24 November 2021, Artificial Intelligence tools can increasingly assist humanity, and all countries should benefit from them. However, they raise several ethical questions. For example, biases will likely rise and cause discrimination, the digital divide, exclusion, and social or economic division. One way to reduce these disasters is to make AI more transparent and understandable, so that the consequences for human dignity, human rights and fundamental freedoms, gender equality, democracy, social, economic, and political processes could be straightforward. This transparent AI is called Explainable AI. (XAI). Among its tools, we can mention Shapley Additive Explanations. (SHAP). SHAP is a technique that facilitates the interpretation of complex model results by focusing on the most critical features and game theory.

The approach is as follows: after selecting a suitable model, we calculate the values of each feature, called SHAP values, by considering all possible groups that can gather these features to obtain a prediction value dedicated to each feature, referred to as the average marginal contribution. Next, we generate a subgroup of features, each being a possible grouping of features for prediction. Then, we calculate the marginal contribution by comparing predictions obtained by including or excluding a specific feature. These marginal contributions are weighted according to the length of the subgroups. From these results, we obtain an average of the marginal contributions for each feature, allowing us to find its Shapley value. These calculated contributions are then summed up, and the total of these Shapley values corresponds to the difference between the model’s final prediction and the baseline prediction. Finally, we can cite the methods to visualize SHAP results: the summary plot, the force plot, the dependence plot, and so on. In the next section, we will see the SHAP results related to the ML predictions in our study using a variant of the summary plot visualization.

## 6. Results and Discussion

### 6.1. Approach

#### 6.1.1. ADHD Children

For the case of ADHD children, we pre-processed the dataset from the NSCH 2022. Then, we divided it into three sets: training, validation, and testing. Subsequently, we applied Machine Learning techniques (LR, DT, KNN, SVM, and MLPC). It turned out that LR was the most suitable algorithm. After explaining these results through SHAP, we noted that excessive laughter/smiling and the number of hours of sleep are the most impactful factors. The following architecture in Figure 1 provides more details about the approach applied:

#### 6.1.2. ADHD Adults

As for the ADHD adults, we took the ADHD|Mental Health dataset from the Kaggle platform and pre-processed it. Then, we divided it into three sets: training, validation, and testing. Subsequently, we applied Machine Learning techniques. (SVM, XGB et MLPC). It turned out that XGB is the most promising algorithm. After explaining these results through SHAP, we noticed that anxiety and depression influence mainly the risk of having ADHD. Figure 2 details the approach applied for this category:

### 6.2. Children

#### 6.2.1. With Feature Selection

According to the results in Table 4, we observe that the LR is the most effective algorithm, achieving an accuracy of 99%, followed by the minimal Loss, Mean Squared Error (MSE), and Mean Absolute Error (MAE) values of 0.001, along with minimal Root Mean Squared Error (RMSE) and Residual Standard Error (RSE) values of 0.011. The MLPC has similar results to LR in Accuracy and Loss and has the best F1-Score with a value of 0.997. Logistic Regression outperforms other ML methods due to its simplicity of interpretation and low risk of overfitting. At the same time, MLPC shows promising results with its ability to learn complex data using fewer layers than other networks like DNN and its training speed.

#### 6.2.2. Without Feature Selection

According to the results in Table 5, the DT has the highest accuracy and F1 score, with a value of 100% and errors reaching 0, except the RMSE, which has a contradictory value of 41.45. On the other hand, we have the MLPC with a lower accuracy of 90.7%, a Loss of 0.092, an MSE of 0.169, and a more reasonable RMSE value of 0.411, followed by an F1-Score of 0.485. Without the feature selection technique, the MLPC remains the most effective ML method due to its characteristics, such as using fewer layers to learn more complex data compared to the DNN and its faster training time.

#### 6.2.3. SHAP Results

Figure 3 below shows the summary chart for the ADHD children:

According to the results of Figure 3 and Table 6, we deduce that the factors that contribute the most to determining a child affected by ADHD are the variable K2Q31C, which describes the severity of ADHD or Attention Deficit Disorder, HOURSLEEP05 representing the sleep hours number, and K6Q72_R, a variable indicating whether the child smiles or laughs. We can see in Figure 3 that, at the beginning, the variable K6Q72_R does not impact the state of the child with ADHD, but as the SHAP value increases, this feature becomes more significant. The higher the value of K6Q72_R is, the more likely the child will have ADHD. Gustafsson HC et al. [85] mentioned children often laugh between 6 and 12 months, and based on the meta-analysis conducted by Zhou R et al. [86], children who laugh more than usual have a higher risk of having ADHD.

### 6.3. Adults

#### 6.3.1. With Feature Selection

The results of Table 7 prove that XGB has the best outcomes, with an accuracy of 100%, the least errors, and the highest F1-Score, reaching a value of 0.987, followed by the SVM algorithm, with an accuracy of 90%, a Loss, MSE, and MAE of 0.028, and an F1-Score of 0.949. XGB outperforms other algorithms due to its ability to combine decision tree models to create a robust model and regularization methods that help prevent overfitting. As for SVM, it is effective in small datasets, maximizing the margin between different classes through its hyperplane and transforming non-linear data into a space to make it linear using kernel functions.

#### 6.3.2. Without Feature Selection

We observe in Table 8 that XGB outperforms the other two ML algorithms across all seven metrics, just like Table 7, which uses feature selection. Compared with the previous table, XGB has an accuracy of 88.4%, with more significant errors, such as loss, MSE, and MAE, with a value of 0.115 and a lower F1-Score of 0.810. The SVM also shows good results, although its values are less significant than those in Table 7, with an accuracy of 84%, a Loss, MSE, and MAE of 0.159, and an F1-Score of 0.765. XGB remains at the top of the ranking thanks to its features, such as the ability to obtain a strong model based on an ensemble of weaker DT models and its L1 and L2 regularizations. The SVM maintains its position due to its advantages, such as its efficient hyperplane for maximizing separation between classes and its kernel functions that make non-linear data linear in space.

#### 6.3.3. SHAP Results

The following Figure 4 displays the summary chart for the ADHD adults:

In agreement with the results in Figure 4 and Table 9, we can see that the factors that most influence ADHD prediction in adults are the bdi1_total (Beck Depression Inventory), the bai1_total (Beck Anxiety Inventory), and questions asked by a specialist, such as those related to mental health issues before university.

The first factor has a value of 35, the second a value of 47, and the third a value of 1. We conclude that the two most significant factors in determining ADHD in an adult are anxiety and depression.

### 6.4. Discussion

The paper aimed to predict Attention Deficit Hyperactivity Disorder (ADHD) for children and adults and understand the most impacting factors. ADHD is a disorder affecting children in many life aspects, such as education, social life, cognitive abilities, and many other difficulties. These complications could persist into their adulthood. Several studies mentioned the challenges children and adults with ADHD are dealing with, such as sleep disorders, high step rates, types of aggression, high caffeine consumption, disorganization at work (Kim WP et al. [6], Meachon EJ et al. [7], Ayasrah MN et al. [8], Cohen A et al. [11] and Durand G et al. [12]), and so on. Others pointed out comorbidities associated with ADHD, for example, epilepsy (Park KJ et al. [9]), anxiety (Gair SL et al. [10], Alarachi A et al., Sacco R et al. [18,19]), alcohol dependence (Pierre-paul L [13]), bipolar and mood disorders (Mishra VC et al. [14]), pragmatic disorders (Even-Simkin E [15]), depression (Borden LA et al. [16], Powell V et al. [17]), and so forth. This paper referenced the prevalence of three comorbidities starting from the childhood: anxiety (Sacco R et al. [19], Alfakeh SA et al. [20],…), depression (Chen J et al. [27], Alaqeel A et al. [28],…) and bipolarity (Wu Y et al. [35])., Mohammadi MR et al. [36],…) as well as their relationship with ADHD, such as environment and genetic factors for the anxiety (Koyuncu A et al. [23]), excessive anxiety for ADHD people (Wu M et al. [26]), stages of puberty for depressive ADHD (Eng AG et al. [33]), chronical diseases like congenital heart disease CHD (Gonzalez VJ et al. [34]), children and adolescent reactions to pharmacological ADHD treatments through mood stabilizers (Pouchon A et al. [40]), and similar decrease in brain regions such as right insula (Long Y et al. [41]). These comorbidities could persist in adulthood. Some research cited adult symptoms, for instance, worry and more maladaptive beliefs for anxious ADHD (O’Rourke SR et al. [42]). Others gave an interest in the prevalence of these comorbidities in adulthood, like 12.8% of depressed ADHD women (Powell V et al. [46]) or sociodemographic factors, including unemployment or unmarried status for ADHD men (Bartoli F et al. [51]). However, therapeutic implications were also mentioned to reduce or alleviate these symptoms. For example, Cognitive Behavioral Therapy (CBT) (León-Barriera R et al. [44]), Mindfulness (Poissant H et al. [48], MORAITI et al. [49]), or mood stabilization (MacDonald L et al. [50]). Additionally, studies have proved evidence that health factors, such as nutrition, have an impact on ADHD. For example, there is a positive correlation between the inattention symptom and seafood or sugar and a negative correlation with healthy foods (Li L et al. [54]), a higher Body Mass Index (Salvat H et al. [56]), or the effectiveness of physical activities for ADHD patients (Xie Y et al. [55]). Others focused on environmental factors, for instance, parenting and children’s education (Claussen AH et al. [61]), pollution (Rosi E et al. [63]), or the COVID-19 pandemic (Hollingdale J et al. [64]). Based on this literature review, and to better understand ADHD, we proposed the following methodology: Firstly, we chose two distinct datasets: the National Survey for Children’s Health (NSCH) dataset for children and the ADHD | Mental Health dataset for adults. Secondly, we pre-processed them by manually removing missing, erroneous values, features having less relevance to ADHD, such as the mother’s age for the children case (Sepehrmanesh Z et al. [69]), or features representing only partial results of a global ADHD score. Thirdly, we split them into train (80% for children and 60% for adults), validation (10% for children and 5% for adults), and test sets (20% for children and 40% for adults). Fourthly, we utilized the Python language and many libraries for the ML models, among them, numpy, scikit-learn, Tensorlfow, and so on. Fifthly, we adjusted the ML models with hyperparameters such as the max_depth or the min_samples_leaf for the Decision Tree (DT) algorithm, kernel or class_weight for the Support Vector Machine (SVM) algorithm, the hidden_layer_sizes or the validation_fraction for the Multi-Layer Perceptron Classifier (MLPC) algorithm. We also used metrics including Accuracy, Mean Squared Error (MSE), F1-Score, and so on. The results showed that, for the children’s case, the Logistic Regression (LR) algorithm was the most suitable among the others, with an accuracy of 99% and achieving a very low error rate of 1.1% for RMSE and RSE. As for the adults, the XGBoost was the most adequate ML method, with an accuracy of 100% and low loss and MSE errors of 0.7%. Finally, to extract the insights that contributed to the prediction of ADHD, we used an XAI method called Shapley Additive Explanations (SHAP). Among the results, the lack of sleep and excessive laughing have a significant impact on ADHD children, while anxiety and depression affect mostly ADHD adults.

As mentioned above, some brain regions may cause emotion dysregulation in patients affected by ADHD. Because of that, they may face many challenges in their social, emotional, or professional life, such as misjudgment, where their laugh could be considered inappropriate, or marginalization, where the ADHD individual deals with negative stereotypes. Another issue could be emotional amplification. The ADHD person who laughs excessively could lose their temper, act more impulsively, and develop comorbidities such as anxiety. The ADHD patient could also lack self-esteem if they are often criticized because of their amplified emotions, leading to severe psychological problems. Their attitude at work could be interpreted as immaturity, reducing their ability to maintain their function. In addition to this, Gustafsson HC et al. and Zhou R et al. [82,83] pointed out that excessive laughter is one of the behaviors that can increase the risk of having ADHD.

Lack of sleep, on the other hand, presents different challenges for the ADHD individual. For example, it can reduce their concentration and attention and increase their irritability and anxiety. Mathur R [22] concluded that 18% of anxious people affected by ADHD lack sleep. The ADHD person may struggle with organization in their professional life, decreasing their work quality and leading them to miss courses or meetings. Moreover, it can intensify mood disorders and result in significant issues such as depression. Subsequently, tiredness, a common symptom of sleep deprivation, can decrease the energy of the concerned patient, impacting their physical abilities. While this study showed great achievements, some improvements could be considered. For example, the explanation of more impairments or the prediction of comorbidities along with ADHD.

## 7. Conclusions and Future Work

To summarize, we began by presenting ADHD, its symptoms: the inattention and hyperactivity/impulsivity, the challenges faced by children and adults such as sleep disorders or caffeine consumption, and some comorbidities like anxiety, mood disorders, epilepsy, depression, bipolar disorder, autism, and so forth. Subsequently, we analyzed three of these comorbidities: anxiety, depression, and bipolar disorder, their prevalence and symptoms, their relationships to ADHD, their potential continuity into adulthood, and some therapeutic implications. Next, we listed the impact of ADHD and these three comorbidities on the health and environment of the affected patients. After that, we proposed a methodology for predicting this disorder in the case of children and adults using two datasets, ML algorithms (LR, DT, KNN, SVM, XGB, and MLPC), and we extracted the most impactful features of ADHD in both cases. More research needs to be conducted to enhance the ADHD people’s health situation in real-life experiences.

## Figures and Tables

**Figure 1 healthcare-13-00155-f001:**
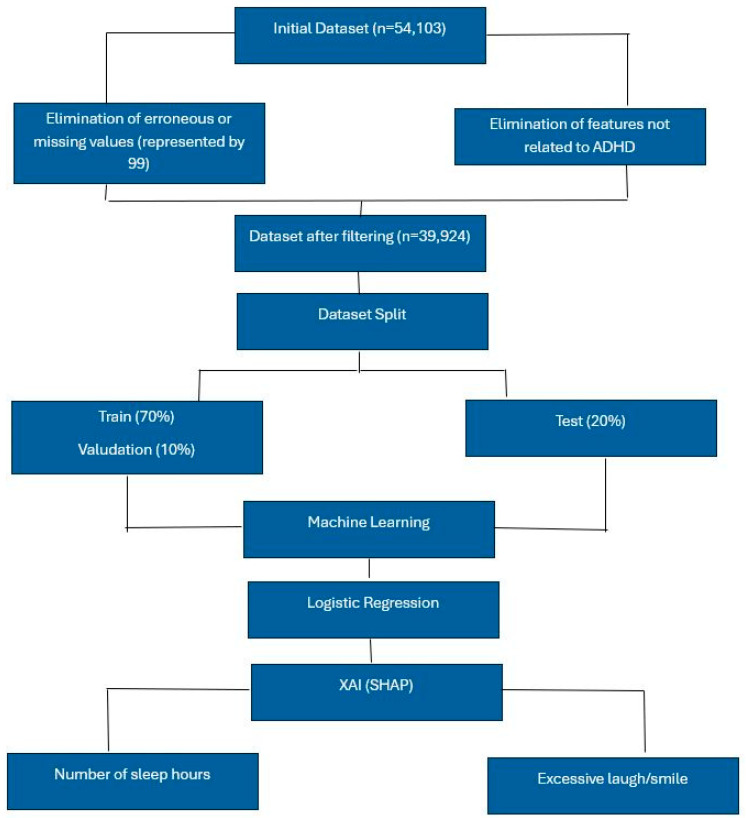
Architecture for ADHD children.

**Figure 2 healthcare-13-00155-f002:**
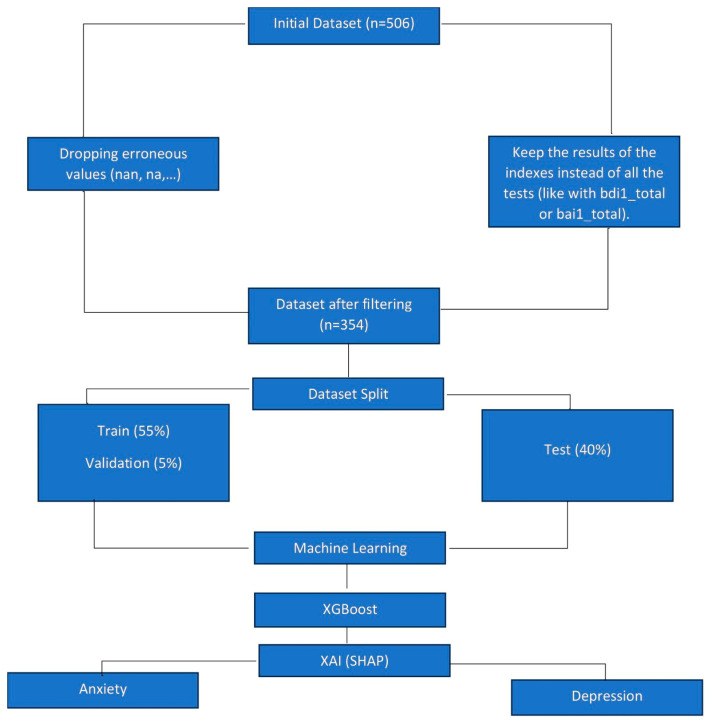
Architecture for ADHD adults.

**Figure 3 healthcare-13-00155-f003:**
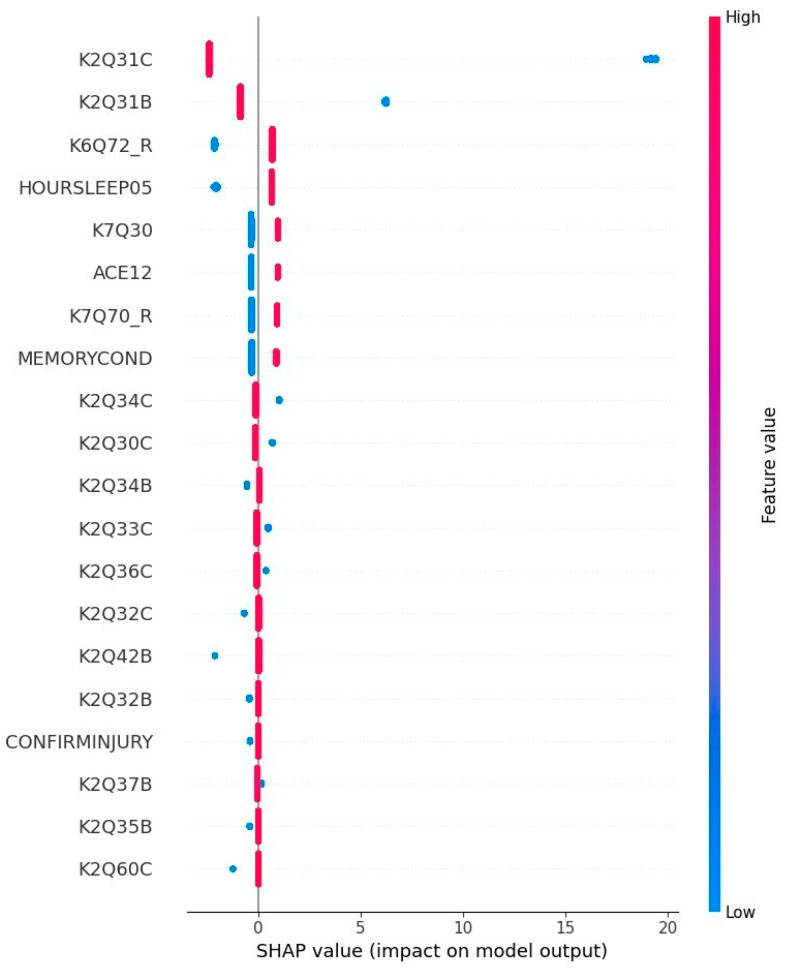
Summary chart for the case of children with ADHD. Notes: K2Q31C: Severity of ADD/ADHD. K2Q31B: ADD/ADHD. Currently, K6Q72_R: whether the child smiles or laughs. HOURSLEEP05: sleep hours number. K7Q30: sports team or sports lessons—past 12 months. ACE12: child experienced—treated unfairly because of their sexual orientation or gender identity. K7Q70_R: argues too much. MEMORYCOND: serious difficulty concentrating, remembering, or making decisions. K2Q34C: behavior problems severity description. K2Q30C: learning disability severity description. K2Q34B: behavior problems. Currently, K2Q33C: Anxiety severity description. K2Q36C: developmental delay severity description. K2Q32C: depression severity description. K2Q42B: epilepsy. Currently, K2Q32B: depression. Currently, CONFIRMINJURY: concussion/brain injury—confirmed injury. K2Q37B: speech disorder. Currently, K2Q35B: autism ASD. Currently, K2Q60C: intellectual disability severity description.

**Figure 4 healthcare-13-00155-f004:**
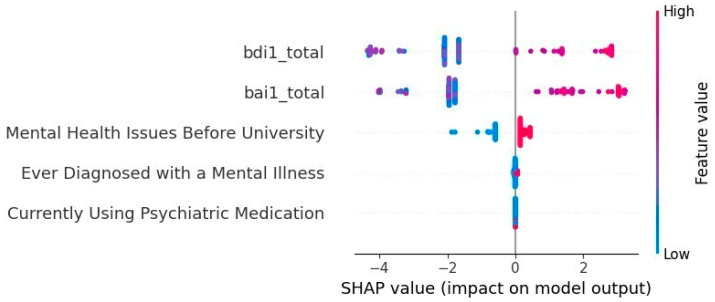
Summary chart for the case of adults with ADHD. Notes: bdi1_total: total result of Beck Depression Inventory; bai1_total: total result of Beck Anxiety Inventory.

**Table 1 healthcare-13-00155-t001:** Name, meaning, and margins of some features of the Dataset on children with ADHD [68].

Features	Meaning	Margins
K2Q42B	Currently epileptic	1: No epileptic 2: Heard about it, but not affected by it 3: Epileptic 95: Children between 0 and 2 years old.
anxiety_22	Children from 3 to 17 years old currently suffering from anxiety	1: Not anxious 2: Heard about it, but not affected by it 3: Anxious 95: Children between 0 and 2 years old
behavior_22	Children from 3 to 17 years old currently having behavioral problems	1: Not having behavioral problems. 2: Heard about it, but not affected by it. 3: Having behavioral problems. 95: Children between 0 and 2 years old.
autism_22	Children from 3 to 17 years old who suffer currently from Autistic Spectrum Disorder	1: Not autistic. 2: Heard about it, but not affected by it. 3: Autistic. 95: Children between 0 and 2 years old.
ADHD_22	Children from 3 to 17 years old who currently have ADD/ADHD	0: Not having ADHD. 1: Having ADHD.

**Table 2 healthcare-13-00155-t002:** Name, meaning, and margins of some characteristics of the Dataset on adults with ADHD.

Features	Meaning	Margins
bdi1_total	Beck Depression Inventory MR Vet al. [71]	1–10: Normal 11–16: Mild depressive disorder 17–20: Low depressive disorder 21–30: Moderate depressive disorder. 31–40: Severe depressive disorder
bai1_total	Beck Anxiety Inventory RECTOR NA et al. [72]	0–7: Mild anxiety disorder 8–15: Low anxiety disorder 16–25: Moderate anxiety disorder 30–63: Severe anxiety disorder
have_you_ever_ experienced_any_ mental_health_difficulties_ or_symptoms_before_starting_ university_e_g_in_primary_or_high_school	Mental Health Issues Before University	0: no 1: yes
have_you_ever_been_ diagnosed_with_a_ mental_illness	Ever Diagnosed with a Mental Illness	0: no 1: yes
are_you_currently_using_prescribed_psychiatric_medication_for_a_mental_illness_or_symptoms_of_one	Currently Using Psychiatric Medication	0: no 1: yes

**Table 3 healthcare-13-00155-t003:** Comparative table of the most used DT algorithms.

Methods	CART	C4.5	CHAID	QUEST
The measure used for input variable collection	Gini index; Twoing criteria	Entropy Info-gain	Chi-square	Chi-square for categorical variables; J-way ANOVA for continuous/ordinal variables
Pruning	Pre-pruning using a single-pass algorithm	Pre-pruning using a single-pass algorithm	Pre-pruning using Chi-square test for independence	Post-pruning
Dependent variable	Categorical/Continuous	Categorical/Continuous	Categorical	Categorical
Input variables	Categorical/Continuous	Categorical/Continuous	Categorical/Continuous	Categorical/Continuous
Split at each node	Binary; Split on linear combinations	Multiple	Multiple	Binary; Split on linear combinations

**Table 4 healthcare-13-00155-t004:** Comparison between metric-based algorithms with feature selection for children with ADHD.

Metrics/Algorithms	LR ^a^	DT ^b^	KNN ^c^	SVM ^d^	MLPC ^e^	
Accuracy	0.999	0.993	0.995	0.986	0.999	
Loss	0.0001	0.006	0.004	0.013	0.0001	
MSE	0.0001	0.006	0.006	0.044	0.001	
MAE	0.0001	0.006	0.004	0.023	0.001	
RMSE	0.011	0.079	0.078	0.209	0.031	
RSE	0.011	0.08	0.078	0.21	0.032	
F1-Score	0.996	0.665	0.866	0.885	0.997	

Notes: ^a^: Logistic Regression; ^b^: Decision Tree; ^c^: K-Nearest Neighbors; ^d^: Support Vector Machine; ^e^: Multi-Layer Perceptron Classifier.

**Table 5 healthcare-13-00155-t005:** Comparison between metric-based algorithms without feature selection for children with ADHD.

Metric/Algorithm	LR ^a^	DT ^b^	KNN ^c^	SVM ^d^	MLPC ^e^
Accuracy	0.7401	1.0	0.836	0.726	0.907
Loss	0.259	0.0	0.163	0.273	0.092
MSE	1534.53	0.0	635.9	1649.59	0.169
MAE	16.49	0.0	6.941	17.714	0.169
RMSE	39.17	41.45	25.21	40.615	0.411
RSE	41.45	0.0	26.68	42.981	0.613
F1-Score	0.306	1.0	0.431	0.21	0.485

Notes: ^a^: Logistic Regression; ^b^: Decision Tree; ^c^: K-Nearest Neighbors; ^d^: Support Vector Machine; ^e^: Multi-Layer Perceptron Classifier.

**Table 6 healthcare-13-00155-t006:** Table of the importance of characteristics for children with ADHD.

Feature	Value
HOURSLEEP05	90.00
K6Q72_R	90.00
MEMORYCOND	2.00
ACE12	2.00
K7Q30	2.00

Notes: HOURSLEEP05: number of hours of sleep. K6Q72_R: whether the child smiles or laughs. MEMORYCOND: severe difficulty in concentrating, remembering, or making decisions. ACE12: child treated unfairly due to their sexual orientation or gender identity. K7Q30: sports team or sports lessons—last 12 months.

**Table 7 healthcare-13-00155-t007:** Comparison between metric-based algorithms with feature selection for adults with ADHD.

Metric/Algorithm	SVM ^a^	XGB ^b^	MLPC ^c^
Accuracy	0.900	1.0	0.882
Loss	0.028	0.007	0.117
MSE	0.028	0.007	0.154
MAE	0.028	0.007	0.154
RMSE	0.167	0.083	0.393
RSE	0.171	0.085	0.402
F1-Score	0.949	0.987	0.692

Notes: ^a^: Support Vector Machine; ^b^: XGBoost; ^c^: Multi-Layer Perceptron Classifier.

**Table 8 healthcare-13-00155-t008:** Comparison between metric-based algorithms without feature selection for adults with ADHD.

Metric/Algorithm	SVM ^a^	XGB ^b^	MLPC ^c^
Accuracy	0.84	0.884	0.818
Loss	0.159	0.115	0.181
MSE	0.159	0.115	0.181
MAE	0.159	0.115	0.181
RMSE	0.399	0.340	0.425
RSE	0.541	0.461	0.577
F1-Score	0.765	0.810	0.450

Notes: ^a^: Support Vector Machine; ^b^: XGBoost; ^c^: Multi-Layer Perceptron Classifier.

**Table 9 healthcare-13-00155-t009:** Table of important characteristics for the case of adults with ADHD.

Feature	Value
Bdi1_total	35.00
Bai1_total	47.00
are_you_currently_using_prescribed_psychiatric_medication _for_a_mental_illness_or_symptoms_of_one	1.00
have_you_ever_been_diagnosed_with_a_mental_illness	1.00

Notes: bdi1_total: total result of Beck Depression Inventory; bai1_total: total result of Beck Anxiety Inventory.

## Data Availability

The original contributions presented in the study are included in the article. Further inquiries can be directed to the corresponding author.

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
