# Peer review of "Explainable Artificial Intelligence for Predicting Attention Deficit Hyperactivity Disorder in Children and Adults"

_healthcare, 2025, doi:10.3390/healthcare13020155_

Round 1

Reviewer 1 Report

Comments and Suggestions for Authors

Thank you for inviting me to review this interesting research topic. Predicting ADHD in children and adults using machine learning and explainable artificial intelligence methods to identify key influencing factors is indeed valuableThis study indeed holds certain reference value. However, I have some concerns and doubts about the details presented in the manuscript. The following are my areas of concern:

1The background section in the introduction is overly lengthy and could be streamlined by removing unnecessary historical details. In Section 3 (comorbidities, relationships with ADHD, continuity, and therapeutic implications), the extensive citations and overly detailed descriptions lack summarization. Additionally, the discussion section is too brief, and I recommend restructuring the manuscript for better balance.

2The data preprocessing section lacks specific details on operations such as handling missing values and feature selection. Please provide additional explanations, such as the criteria for removing certain features.

3The description of the dataset and the structure diagrams (Figures 1 and 2) are inconsistent. Please revise for accuracy and consistency.

4The results and discussion section focuses too heavily on the performance of the machine learning models and lacks an in-depth discussion of ADHD. I recommend combining the model results with a thorough analysis of the influencing factors of ADHD and their practical implications.

5Some sentences need refinement for precision and clarity. For instance, the phrase exacerbate a few disorders could be improved to exacerbate disorders such as....

Comments on the Quality of English Language

The language should be improved.

Author Response

Original Manuscript ID: Healthcare-3381872

Original Article Title: Explainable Artificial Intelligence for predicting Attention Deficit Hyperactivity Disorder in children and adults

To: MDPI Healthcare

Re: Response to reviewers

Dear Editor, Dear Reviewer,

We appreciate the chance to revise and resubmit our work to address the concerns provided by the reviewers.

We have uploaded our detailed response to the remarks (the response to reviewers), along with a revised manuscript that includes highlighted revisions in yellow.

Best regards,

Zineb Namasse, Mohamed Tabaa , Zineb Hidila, Samar Mouchawrab

+----------------------------------------------------------------------------------------------------------------+

Review 1:

+----------------------------------------------------------------------------------------------------------------+

Remark 1: The background section in the introduction is overly lengthy and could be streamlined by removing unnecessary historical details. In Section 3 (comorbidities, relationships with ADHD, continuity, and therapeutic implications), the extensive citations and overly detailed descriptions lack summarization. Additionally, the discussion section is too brief, and I recommend restructuring the manuscript for better balance.

Answer 1: Thank you for providing me this remark. We reduced the Background Section Length. In the Section 3, the descriptions are summarized. For example, from line 97 to line 99, “Several authors reported anxiety rates of different countries such as England, Australia (nationwide), Turkey (Izmir) and Lithuania (nationwide), where anxiety had the highest prevalence at 7.9% (Sacco R et al. [19])…”. The Discussion Section contains now more details.

+----------------------------------------------------------------------------------------------------------------+

Remark 2: The data preprocessing section lacks specific details on operations such as handling missing values and feature selection. Please provide additional explanations, such as the criteria for removing certain features.

Answer 2: The data preprocessing section contains now more details on feature selection. For instance, in the sub sub-section “Dataset for children”, from line 288 to line 295, “We then selected features by removing manually erroneous or missing values represented by 99, features not related to children, like A1_DEPLSTAT/A2_DEPLSTAT, which concern adults, or MOMAGE, representing the mother's age. According to an article by Sepehrmanesh Z et al. [69] on the links between breastfeeding and ADHD, along with previous results (Mimouni – Bloch et al., 2013; Shamberger, 2012; Stadler et al., 2015; 2017; Visser et al., 2013, and so forth.), the mother's age during pregnancy does not play a significant role in the cases of ADHD in comparison to non-ADHD cases”. As for the sub sub-section “Dataset for adults”, from line 313 to line 316, “We proceeded as follows: we started by manually removing the incorrect or erroneous values represented by n/a, na..., indexes representing only a part of the total number of an index, so we kept instead the total numbers (bai1_total, bdi1_total,).”.

+----------------------------------------------------------------------------------------------------------------+

Remark 3: The description of the dataset and the structure diagrams (Figures 1 and 2) are inconsistent. Please revise for accuracy and consistency.

Answer 3: After checking the figures 1 and 2, the dataset descriptions were corrected. In the sub sub-section “Dataset for children”, in line 303, “70% for training” instead of 80%. And in the sub sub-section “Dataset for adults”, in line 320, “55% were dedicated to training” instead of 60%.

+----------------------------------------------------------------------------------------------------------------+

Remark 4: The results and discussion section focuses too heavily on the performance of the machine learning models and lacks an in-depth discussion of ADHD. I recommend combining the model results with a thorough analysis of the influencing factors of ADHD and their practical implications.

Answer 4: In the Discussion section, we added an analysis of the ADHD factors and their implications from line 723 to line 743. For example, “As mentioned above, some brain regions may cause emotion dysregulation in patients affected by ADHD. Because of that, they may face many challenges in their social, emotional or professional life such as misjudgement, where their laugh could be considered inappropriate or marginalization, where the ADHD individual deals with negative stereotypes. Another issue could be the emotion amplification…”

+----------------------------------------------------------------------------------------------------------------+

Remark 5: Some sentences need refinement for precision and clarity. For instance, the phrase “exacerbate a few disorders” could be improved to “exacerbate disorders such as....”

Answer 5: Some sentences were refined. As an example, from line 62 to 63 in the Introduction section, instead of “among the risks to mental health that could lead to mental disorders such as depression”, “among the mental health problems that cause mental disorders such as depression…”

Reviewer 2 Report

Comments and Suggestions for Authors

Dear Authors,

After reviewing your manuscript “Explainable Artificial Intelligence for predicting Attention Deficit Hyperactivity Disorder in children and adults”, some aspects came out. Firstly, the idea addressed in your paper is relevant, the prediction of ADHD using machine learning techniques and its implications for both children and adults.

The title – may be to include in the title also, if you think is appropriate – the comorbidities, for example, “Explainable Artificial Intelligence for predicting Attention Deficit Hyperactivity Disorder and its comorbidities in children and adults”.

Abstract – I would remove from the title “etc…” Also, clearly state the article aim, the following sentence “In this article, we present the ADHD by citing certain physicists and doctors interested in this disease, its main symptoms, challenges faced by children and adults such as epilepsy and depression. Then, we explore deeply three the comorbidities of this disease for both categories: anxiety, depression and bipolar disorders, we explain the continuity possibility of these impairments in adulthood and list a few therapeutic implications” is too vague. Avoid vague phrases like "certain physicists and doctors interested in this disease" and replace them with precise references to relevant research or fields of study.

Provide more detail on the datasets used (e.g., size, source, or population demographics) to add context to the methodology.

Introduction

In the first sentence “According to [1]….”, consider mentioning the name and place the reference number after it, for example, “According to WHO [1]….”

The second paragraph of the introduction, when you introduce ADHD – provide some definitions of the disorders, citing well knowing specialists and experts in the field or citing DSM V, for example I will move the fourth paragraph earlier in the paper, making it the second paragraph. In my view the history of ADHD it is not relevant as your paper is focusing on the diagnosis.

Second page, line 76 – check the consistency, you write Working Memory with capital letters, while other characteristics are not, use small letters for working memory too. Also avoid etc. …

Line 79 – you can not start your sentence with “[4] mentions that among children with ADHD, boys are more impulsive while girls 79 tend to be more inattentive”. Provide the name of the author “Desseilles et al. [4 ] mention that among children with ADHD, boys are more impulsive while girls 79 tend to be more inattentive].

Line 92- 93 The purpose of the article – make it last sentence of your introduction section.

Your section 2 Background - is too long, you have very long paragraphs, too much details and there is a lack of critical analysis and a synthesis of the data you are providing. I would focus on some main characteristics of ADHD, not entering into too many details and would discuss more deeply the challenges of ADHD and its comorbidities diagnosis, if AI was used, is used and how it can predict ADHD and how professionals could use it. Also, to provide some studies, if there are any on this topic.

Identify more clearly the research gap.

Half of your paper is on introduction and literature review, and it should not exceed 1/3.

 Methods and materials.

Please explain in more details, your study methodology – section 5.2. Methods, which research methods did you use, the rational of the selected methods, before discussing the choice of AI models.

Details of models used – I do not have competences to review this part.  

Section 6 – Results and Discussion

You present only results here, you could name it only results or eliminate section 7 – discussion.

The discussion part should be one of the strongest parts of your research, however, it is not consistent, you should contribute more to this part. Begin the discussion section by reminding the readers your research aim. Clarify how the key predictors (e.g., sleep hours, excessive smiling/laughter, anxiety, depression) were selected. Discuss whether any features were excluded due to collinearity or lack of relevance. You could describe the validation process in detail and address how the models were tuned to achieve their respective accuracy. Specify how explainable AI can help predict ADHD and interpret the results.

Also, I would add a limitation section, where you could add some limitations of your study.

Kind regards,

Author Response

Original Manuscript ID: Healthcare-3381872

Original Article Title: Explainable Artificial Intelligence for predicting Attention Deficit Hyperactivity Disorder in children and adults

To: MDPI Healthcare

Re: Response to reviewers

Dear Editor, Dear Reviewer,

We appreciate the chance to revise and resubmit our work to address the concerns provided by the reviewers.

We have uploaded our detailed response to the remarks (the response to reviewers), along with a revised manuscript that includes highlighted revisions in yellow.

Best regards,

Zineb Namasse, Mohamed Tabaa , Zineb Hidila, Samar Mouchawrab

+----------------------------------------------------------------------------------------------------------------+

Review 2:

+----------------------------------------------------------------------------------------------------------------+

Remark 1: The title – may be to include in the title also, if you think is appropriate – the comorbidities, for example, “Explainable Artificial Intelligence for predicting Attention Deficit Hyperactivity Disorder and its comorbidities in children and adults”.

Answer 1:  Thank you for your remark. However, the main objective of this paper is divided into 2 parts: Firstly, the prediction of only ADHD using ML methods. Secondly, using XAI to understand the factors having an impact on ADHD and among them, comorbidities.

+----------------------------------------------------------------------------------------------------------------+

Remark 2: Abstract – I would remove from the title “etc…” Also, clearly state the article aim, the following sentence “In this article, we present the ADHD by citing certain physicists and doctors interested in this disease, its main symptoms, challenges faced by children and adults such as epilepsy and depression. Then, we explore deeply three the comorbidities of this disease for both categories: anxiety, depression and bipolar disorders, we explain the continuity possibility of these impairments in adulthood and list a few therapeutic implications” is too vague. Avoid vague phrases like "certain physicists and doctors interested in this disease" and replace them with precise references to relevant research or fields of study.

Provide more detail on the datasets used (e.g., size, source, or population demographics) to add context to the methodology.

Answer 2: The “etc…” terms are removed from the abstract. The article aim is less vague. From line 13 to line 16, “we introduce the main symptoms and challenges ADHD poses for children and adults such as epilepsy and depression. Then, we present results of existing research on three ADHD comorbidities: anxiety, depression and bipolar disorder…”

+----------------------------------------------------------------------------------------------------------------+

Remark 3: Introduction

In the first sentence “According to [1]….”, consider mentioning the name and place the reference number after it, for example, “According to WHO [1]….”

Answer 3: The reference is added after the number. In line 32, “According to the World Health Organization (WHO) [1]..”

+----------------------------------------------------------------------------------------------------------------+

Remark 4: The second paragraph of the introduction, when you introduce ADHD – provide some definitions of the disorders, citing well knowing specialists and experts in the field or citing DSM V, for example I will move the fourth paragraph earlier in the paper, making it the second paragraph. In my view the history of ADHD it is not relevant as your paper is focusing on the diagnosis.

Answer 4: We placed the fourth paragraph in the second one. From line 37 to line 40, “ADHD is a neurodevelopmental disorder that affects 3-10% of children and 4-5% of adults (Desseilles et al. [2]). The main symptoms are the Attention Deficit and the Hyperactivity/Impulsivity disorder (Simmons RW et al. [3] and the Diagnostic and Statistical Manual of Mental Disorders, 5th edition (DSM-V)).”. The ADHD history is removed.

+----------------------------------------------------------------------------------------------------------------+

Remark 5: Second page, line 76 – check the consistency, you write Working Memory with capital letters, while other characteristics are not, use small letters for working memory too. Also avoid etc. …

Answer 5: the “ect..” term is removed from the entire paper. The words “working memory” has now small letters. In line 58, which was line 76 before, “working memory and others…”.

+----------------------------------------------------------------------------------------------------------------+

Remark 6: Line 79 – you can not start your sentence with “[4] mentions that among children with ADHD, boys are more impulsive while girls 79 tend to be more inattentive”. Provide the name of the author “Desseilles et al. [4 ] mention that among children with ADHD, boys are more impulsive while girls 79 tend to be more inattentive]

Answer 6: Names of authors are now provided before each reference number. For example, from line 60 (line 79 before) to line 62, “Desseilles et al. [2] mentioned that among children with ADHD, boys are more impulsive while girls tend to be more inattentive…”.

+----------------------------------------------------------------------------------------------------------------+

Remark 7: Line 92- 93 The purpose of the article – make it last sentence of your introduction section.

Answer 7: The purpose of the article is now mentioned in the last sentence of the Introduction section. From line 73 (before 92) to line 75, “This paper aims to explore how Explainable AI and Machine Learning techniques can contribute to the prediction of ADHD and the understanding of associated comorbidities in children and adults…”.

+----------------------------------------------------------------------------------------------------------------+

Remark 8: Your section 2 Background - is too long, you have very long paragraphs, too much details and there is a lack of critical analysis and a synthesis of the data you are providing. I would focus on some main characteristics of ADHD, not entering into too many details and would discuss more deeply the challenges of ADHD and its comorbidities diagnosis, if AI was used, is used and how it can predict ADHD and how professionals could use it. Also, to provide some studies, if there are any on this topic Identify more clearly the research gap. Half of your paper is on introduction and literature review, and it should not exceed 1/3.

Answer 8: We reduced the Background section size and the Third section size.

+----------------------------------------------------------------------------------------------------------------+

Remark 9: Please explain in more details, your study methodology – section 5.2. Methods, which research methods did you use, the rational of the selected methods, before discussing the choice of AI models.

Answer 9: We explained the methodology in the sub section Methods. From line 331 to line 336, “we proposed the following methodology. We started by handling missing, erroneous values and features not related to ADHD in children and adults in both datasets. Then, we split them into train, validation and test sets. After that, we employed Machine Learning (ML) methods to predict ADHD in children and adults and Explainable Artificial Intelligence (XAI) to reveal the features that contributed, positively or negatively, to the ADHD prediction.”

+----------------------------------------------------------------------------------------------------------------+

Remark 10: You present only results here, you could name it only results or eliminate section 7 – discussion. The discussion part should be one of the strongest parts of your research, however, it is not consistent, you should contribute more to this part. Begin the discussion section by reminding the readers your research aim. Clarify how the key predictors (e.g., sleep hours, excessive smiling/laughter, anxiety, depression) were selected. Discuss whether any features were excluded due to collinearity or lack of relevance. You could describe the validation process in detail and address how the models were tuned to achieve their respective accuracy. Specify how explainable AI can help predict ADHD and interpret the results.

Answer 10:  The aim of the paper is now added to the Discussion sub section. From line 666 to line 667, “The aim of the paper was to predict the Attention Deficit Hyperactivity Disorder (ADHD) for children and adults and understand the most impacting factors.”. We explained the reasons for features exclusion. For instance, from line 704 to line 706, “Secondly, we pre-processed them by manually removing missing, erroneous values, features having less relevance to ADHD, such as the mother’s age for the children case (Sepehrmanesh Z et al. [69])…”. We added further explanation on the impacting factors on the ADHD in the Discussion sub section. For instance, from line 723 to line 727, “As mentioned above, some brain regions may cause emotion dysregulation in patients affected by ADHD. Because of that, they may face many challenges in their social, emotional or professional life such as misjudgment, where their laugh could be considered inappropriate or marginalization, where the ADHD individual deals with negative stereotypes…”.

+----------------------------------------------------------------------------------------------------------------+

Remark 11: Also, I would add a limitation section, where you could add some limitations of your study.

Answer 11: The Limitation section is now added to the paper, right after the discussion sub section.

Round 2

Reviewer 1 Report

Comments and Suggestions for Authors

Thank you. All my concerns have been addressed.

Comments on the Quality of English Language

The language should be improved. Thank you.

Author Response

Original Manuscript ID: Healthcare-3381872

Original Article Title: Explainable Artificial Intelligence for predicting Attention Deficit Hyperactivity Disorder in children and adults

To: MDPI Healthcare

Re: Response to reviewers

Dear Editor, Dear Reviewer,

We appreciate the chance to revise and resubmit our work to address the concerns provided by the reviewers.

We have uploaded our detailed response to the remarks (the response to reviewers), along with a revised manuscript that includes highlighted revisions in red.

Best regards,

Zineb Namasse, Mohamed Tabaa , Zineb Hidila, Samar Mouchawrab

+----------------------------------------------------------------------------------------------------------------+

Review 1:

+----------------------------------------------------------------------------------------------------------------+

Remark 1: The language should be improved. Thank you.

Answer 1: Thank you for providing me this remark. The langage is improved

Reviewer 2 Report

Comments and Suggestions for Authors

Dear authors, 

Thank you for considering the recommendations made, your manuscript has been improved. 

However some minor issues should be discussed before proceeding with the publication process. 

1 . In the abstract - the sentence added, in the objective section is not clear, it does not sound as an objective, please revise it. 

Best, 

Author Response

Original Manuscript ID: Healthcare-3381872

Original Article Title: Explainable Artificial Intelligence for predicting Attention Deficit Hyperactivity Disorder in children and adults

To: MDPI Healthcare

Re: Response to reviewers

Dear Editor, Dear Reviewer,

We appreciate the chance to revise and resubmit our work to address the concerns provided by the reviewers.

We have uploaded our detailed response to the remarks (the response to reviewers), along with a revised manuscript that includes highlighted revisions in red.

Best regards,

Zineb Namasse, Mohamed Tabaa , Zineb Hidila, Samar Mouchawrab

+----------------------------------------------------------------------------------------------------------------+

Review 2:

+----------------------------------------------------------------------------------------------------------------+

Remark 1:  In the abstract - the sentence added, in the objective section is not clear, it does not sound as an objective, please revise it. 

Answer 1:  Thank you for your remark. The objective in the abstract is now clear. From line 14 to line 15, “This paper aims to predict ADHD in children and adults and explain the main factors impacting this disorder”.